# Gene Expression Profiling of *Corpus luteum* Reveals Important Insights about Early Pregnancy in Domestic Sheep

**DOI:** 10.3390/genes11040415

**Published:** 2020-04-10

**Authors:** Kisun Pokharel, Jaana Peippo, Melak Weldenegodguad, Mervi Honkatukia, Meng-Hua Li, Juha Kantanen

**Affiliations:** 1Natural Resources, Natural Resources Institute Finland (Luke), 31600 Jokioinen, Finland; kisun.pokharel@luke.fi (K.P.); melak.weldenegodguad@luke.fi (M.W.); 2Production Systems, Natural Resources Institute Finland (Luke), 31600 Jokioinen, Finland; jaana.peippo@luke.fi; 3NordGen—The Nordic Genetic Resources Center, 1432 Ås, Norway; mervi.honkatukia@nordgen.org; 4College of Animal Science and Technology, China Agricultural University, Beijing 100193, China

**Keywords:** Finnsheep, Texel, progesterone, preimplantation, embryonic diapause

## Abstract

The majority of pregnancy loss in ruminants occurs during the preimplantation stage, which is thus the most critical period determining reproductive success. Here, we performed a comparative transcriptome study by sequencing total mRNA from *corpus luteum* (CL) collected during the preimplantation stage of pregnancy in Finnsheep, Texel and F1 crosses. A total of 21,287 genes were expressed in our data. Highly expressed autosomal genes in the CL were associated with biological processes such as progesterone formation (*STAR*, *CYP11A1*, and *HSD3B1*) and embryo implantation (e.g., *TIMP1*, *TIMP2* and *TCTP*). Among the list of differentially expressed genes, sialic acid-binding immunoglobulin (Ig)-like lectins (*SIGLEC3*, *SIGLEC14*, *SIGLEC8*), ribosomal proteins (*RPL17, RPL34, RPS3A*, *MRPS33*) and chemokines (*CCL5*, CCL24, *CXCL13*, *CXCL9*) were upregulated in Finnsheep, while four multidrug resistance-associated proteins (MRPs) were upregulated in Texel ewes. A total of 17 known genes and two uncharacterized non-coding RNAs (ncRNAs) were differentially expressed in breed-wise comparisons owing to the flushing diet effect. The significantly upregulated *TXNL1* gene indicated potential for embryonic diapause in Finnsheep and F1. Moreover, we report, for the first time in any species, several genes that are active in the CL during early pregnancy (including *TXNL1*, *SIGLEC14*, *SIGLEC8*, *MRP4,* and *CA5A*).

## 1. Introduction

Litter size, a key determinant for the profitability of sheep production systems, is highly dependent on ovulation rate and embryo development in the uterus. Earlier studies have shown that the trait of high prolificacy can result due to the action of either a single gene with a major effect, as in the Chinese Hu, Boorola Merino, Lacaune and small-tailed Han breeds [1,2,3,4,5,6], or different sets of genes, as in the Finnsheep and Romanov breeds [7,8]. The native Finnsheep, one of the most highly prolific breeds, has been exported to more than 40 countries to improve local breeds [9]. In recent years, a FecG^F^ (V371M) mutation in gene *GDF9* has been identified to be strongly associated with litter size in Finnsheep and breeds such as the Norwegian White Sheep, Cambridge and Belclare breeds, which were developed using Finnsheep [10,11,12,13].

The success of pregnancy establishment in sheep and other domestic ruminants is determined at the preimplantation stage and involves coordination among pregnancy recognition, implantation and placentation, in which the *corpus luteum* (CL) and endometrium play vital roles [14,15,16]. The preimplantation stage of pregnancy is the most critical period in determining the litter size because of the high embryo mortality during this period. It has been shown that most embryonic deaths occur before day 18 of pregnancy in sheep [17,18,19]. Flushing (elevated levels of feed) is known to improve blastocyst yield and embryo survival [20,21]. However, due to the biological complexity of the process and to technical difficulties, embryo implantation is still not well understood. The CL is an endocrine structure whose main function is to synthesize and secrete the hormone progesterone. Progesterone production is essential for the establishment of pregnancy. However, if pregnancy is not established, the CL will regress as a result of luteolysis, and a new cycle will begin.

The whole-transcriptome profiling approach enables a deeper understanding of the functions of the CL, which may allow the identification of genes and markers that are differentially expressed, for example, between breeds showing different litter size phenotypes. Although most of the studies associated with early pregnancy have been performed in sheep [14,15,22,23,24], only a few studies have applied transcriptomic approaches to the CL. A microarray-based transcriptomic study conducted identified a number of genes regulated by progesterone (from the CL) and interferon tau (*IFNT*; from the conceptus) in pregnant vs uterine gland knockout (UGKO) ewes [25]. In a more comprehensive study, transcriptome analysis of uterine epithelial cells during the peri-implantation period of pregnancy identified various regulatory pathways and biological processes in sheep [26]. Moore et al. (2016) combined gene expression data with genome-wide association studies (GWASs) to understand the roles of CL and endometrium transcriptomes in dairy cattle fertility [27]. Another study identified differentially expressed genes (DEGs) between Day 4 and Day 11 in the CL in cattle [28]. Though these studies have certainly enhanced our understanding of the roles of the CL during early pregnancy and in ruminant fertility in general, none of these studies conducted specific comparisons between breeds with different reproductive potential. Thus, in this study, a comparison of transcriptome profiles between two breeds (high prolific Finnsheep and low-prolific Texel) was conducted to provide insight into the differences in developmental events in early pregnancy between the breeds. The influence of diet during early pregnancy was assessed by keeping half of the ewes on the flushing diet. Here, the main goal of this study was to build a global picture of transcriptional complexity in Cl and examine differences in developmental profiles during early pregnancy in sheep breeds showing contrasting fertility phenotypes. Thus, this study has relevance to sheep breeding towards achieving improved reproductive capacity.

## 2. Materials and Methods 

### 2.1. Experimental Design

All procedures for the experiment and sheep sampling were approved by the Southern Finland Animal Experiment Committee (approval no. ESAVI/5027/04.10.03/2012). The animals were kept at Pusa Farm in Urjala, located in the province of Western Finland, during the experimental period. A total of 31 ewes representing three breed groups (Finnsheep (n = 11), Texel (n = 11) and F1 crosses (n = 9) were included in the main experiment (please note that only 18 of the 31 ewes have been included in this study) of which approximately half of the ewes from each breed were kept on the flushing diet. Analyses were conducted for two different time points during the establishment of pregnancy: the follicular growth phase [13] and early pregnancy prior to implantation (current study). After ovary removal, the ewes were mated using two Finnsheep rams, and the pregnant ewes were slaughtered during the preimplantation phase of the pregnancy when the embryos were estimated to be one to three weeks old (Appendix A). At the slaughterhouse, a set of tissue samples (the pituitary gland, a CL, oviductal and uterine epithelial cells, and preimplantation embryos) were collected and stored in RNAlater reagent (Ambion/Qiagen, Valencia, CA, USA) following the manufacturer’s instructions. Of the collected tissue samples, CL was subjected to the current study. One of the CLs was dissected from each ovary. For the present study, and particularly for the RNA-Seq of the CL, six ewes each from the Finnsheep, Texel and F1 cross groups were included. Therefore, out of 31 ewes that were originally included in the main experiment, only 18 have been considered here. The experimental design was described in more detail in an earlier study [13].

### 2.2. Library Preparation and Sequencing

RNA was extracted from the tissues using an RNeasy Plus Mini Kit (Qiagen, Valencia, CA, USA) following the manufacturer’s protocol. The details on RNA extraction have been described previously [13]. RNA quality (RNA concentration and RNA integrity number) was measured using a Bioanalyzer 2100 (Agilent Technologies, Waldbronn, Germany) before sending the samples to the Finnish Functional Genomics Center, Turku, Finland, where library preparation and sequencing were performed. RNA libraries were prepared according to the Illumina TruSeq® Stranded mRNA Sample Preparation Guide (part # 15031047) which included poly-A selection step. Unique Illumina TruSeq indexing adapters were ligated to each sample during an adapter ligation step to enable pooling of multiple samples into one flow cell lane. The quality and concentrations of the libraries were assessed with an Agilent Bioanalyzer 2100 (Agilent Technologies Inc., Santa Clara, CA, USA) and by Qubit® Fluorometric Quantitation (Thermo Fisher Scientific, Waltham, MA, USA), respectively. All samples were normalized and pooled for automated cluster preparation at an Illumina cBot (Illumina Inc., San Diego, CA, USA) station. High-quality libraries of mRNA were sequenced with an Illumina HiSeq 2000 (Illumina Inc., San Diego, CA, USA) instrument using paired-end (2 × 100 bp) sequencing strategy.

### 2.3. Data Preprocessing and Mapping

The raw reads were assessed for errors and the presence of adapters using FastQC v0.11.6 [29]. As we noticed the presence of adapters, Trim Galore v0.5.0 [30,31] was used to remove the adapters and low-quality reads and bases. Clean RNA-Seq reads were aligned to the latest sheep reference genome using STAR v2.6.1a [32]. The reference genome (oar_rambouillet_v1.0, BioProject: PRJNA414087) and annotation (NCBI *Ovis aries* Annotation Release 103) were used to construct “Star genome” prior to mapping step. In order to facilitate the differential expression analysis, the “*--quantMode GeneCounts*” option was included in the STAR mapping command. Alternatively, we performed transcript quantification under the quasi-mapping-based mode in Salmon v1.1.0 [33] transcriptome index built using oar_rambouillet_v1.0 transcriptome to get TPM (transcripts per million reads) value of expressed genes.

### 2.4. Differential Gene Expression Analysis

The raw counts quantified as part of STAR mapping were considered for gene expression analysis whereas the Salmon-based transcript estimates were summarized to gene level estimates to prepare a table of genes with their abundance using tximport v1.12.3 [34] Bioconductor package. Prior to gene level summarization, a customized “*tx2gene*” data frame was created using the annotation (*.gtf file*) of Rambouillet v1.0 transcriptome. We used DESeq2 [35] for differential gene expression analysis. Transcripts with less than 5 read counts were discarded and technical replicates (samples representing same animal) were collapsed/merged before running the DESeq command. Pairwise differential expression analysis was performed between Finnsheep, Texel and F1 crosses. Differentially expressed genes with adjusted p-value of 0.05 (*padj < 0.05*) and absolute log2(fold change) of 1.5 (*abs(log2FoldChange) > 1.5*) were regarded as significant in this study. In DESeq2, Benjamini and Hochberg’s method is used for estimating the adjusted p-values. For identifying a subset of genes potentially differentially expressed due to flushing diet effect, we employed a separate design by including diet as a secondary factor. The statistical analyses were performed in R v3.6.0 [36]. 

### 2.5. Functional Analysis of Differentially Expressed Genes

The ClueGO v2.5.5 [37] plugin in Cytoscape v3.7.0 [38] was employed for gene functional analysis. Prior to performing the analyses, we downloaded the latest versions of the Kyoto Encyclopedia of Genes and Genomes (KEGG) pathways and Gene Ontology (GO) terms. The enrichment analysis was based on a one-sided hypergeometric test with the Benjamini–Hochberg correction method. We used a custom reference set that included a list of all the expressed genes in our data. We also modified the default GO and pathway selection criteria in such a way that a minimum of three genes and four percent of genes from a given GO or KEGG pathway should be present in the query list. Furthermore, GO terms with a minimum level of three and a maximum level of eight were retained. Finally, GO terms and/or KEGG pathways sharing similar genes were grouped together using kappa statistics with kappa score threshold of 0.4.

## 3. Results and Discussion

### 3.1. Phenotypic Observations

After removal of the remaining ovary, we counted the number of CLs visually in each animal. With an average of 4.1, Finnsheep had the highest number of CLs, whereas Texel had an average of 1.7 CLs (Table 1, Appendix A). F1 showed phenotypes closer to those of Finnsheep than those of Texel, having 3.75 CLs on average (Appendix A); this was unsurprising, as we observed a similar pattern in an earlier study [13]. We did not observe more than two CLs in the Texel or fewer than three CLs in Finnsheep or F1. Similarly, on average, Finnsheep had the highest number of embryos (n = 2.6), followed by F1 crosses (n = 1.8) and Texel (n = 1.5). Interestingly, the embryo survival rate in Texel was highest, where 1.5 embryos were present from 1.7 CLs (88%) on average. On the other hand, Finnsheep (63%) and F1 cross (48%) had a remarkably low embryo survival rate. While these findings are based on fewer animals, the results are in agreement with earlier studies [39,40] where typically higher litter size is associated with higher embryo mortality and vice versa. It would be of great interest to determine if productivity follows the same pattern in F2 (i.e., F1 × F1) crosses, backcrosses and presumably also in a reciprocal cross.

### 3.2. RNA-Seq Data

From the 42 libraries (21 from each tissue, including three technical replicates), around 1104 million (M) raw reads were sequenced, of which 1094 M clean reads were retained after trimming (Table 2). The summary statistics from Trim Galore revealed that up to 3.6% of the reads were trimmed, with reverse-strand reads having a comparatively higher percentage of trimmed bases. However, the percentage of reads that were excluded for being shorter than 18 bp was always less than 1% across all samples. More than 80% of the reads from all samples were aligned to the Rambouillet reference genome and transcriptome using STAR and Salmon, respectively. The raw RNA-Seq data (Fastq files) from this study are available in European Nucleotide Archive (ENA) under project accession code PRJEB32852. The details of sample summary including ENA accession code of individual samples are available in Table 2.

### 3.3. Gene Expression in the CL 

STAR-based alignment revealed 23,327 genes and pseudogenes expressed in the whole data set which makes up approximately 70% of the list (n = 33,372) available for the recent Rambouillet reference genome. Further grouping of the expressed genes showed that the most genes (n = 21,052) were expressed in Finnsheep followed by Texel (n = 20,957) and F1 crosses (n = 20,928). The cumulative difference in the number of genes in different samples and breeds might be due to transcriptional noise. As shown in Figure 1, the highest number of breed-specific genes was found in Finnsheep (n = 399), followed by Texel (n = 363) and F1 crosses (n = 303). In a pairwise comparison, based on overall gene expression, Finnsheep and F1 crosses shared a higher number of genes (n = 338) than other pairs, indicating the closer relatedness of F1 crosses to Finnsheep (Figure 1). As indicated by the principal component analysis (PCA), we did not observe any breed-specific clusters in CL and this was also the case in our earlier ovarian transcriptome study [13]. Alternatively, Salmon-based quantification revealed 27,072 genes expressed in the CL, of which 18,463 had TPM greater than 0.1 (Appendix A).

### 3.4. Highly Expressed Genes

To obtain an overview of the most abundant genes in the tissue, we selected the top 25 genes expressed in the CL (Table 3) derived from Salmon quantification. We noticed that 10 out of the top 25 genes were ribosomal proteins. Moreover, the top expressed genes also appeared to play substantial roles during the preimplantation stage. Steroidogenic acute regulatory protein (*STAR*), the second most highly expressed gene, plays an important role in mediating the transfer of cholesterol to sites of steroid production [41,42]. Post ovulation, the expression of the majority of genes associated with progesterone synthesis starts to increase and peaks around the late luteal phase, when the CL has fully matured [43,44,45]. *STAR*, together with the cytochrome P450 side chain cleavage (P450cc) complex and 3b-hydroxysteroid dehydrogenase/delta5 delta4-isomerase (*HSD3B1*), are the three most important actors involved in progesterone biosynthesis. *STAR* is involved in transporting free cholesterol to the inner mitochondrial membrane. The P450cc complex, composed of a cholesterol side chain cleavage enzyme (*CYP11A1*), ferredoxin reductase (*FDXR*) and ferredoxin (*FDX1*), converts the newly arrived cholesterol into pregnenolone [46]. Finally, *HSD3B1* helps in converting pregnenolone to progesterone [45,47,48,49]. Three of these major genes involved in progesterone synthesis (*STAR, HSD3B1,* and *HSD3B1*) were ranked among the top 25 most highly expressed autosomal genes, while *FDX1* (TPM = 972.8) and *FDXR* (TPM = 221.5) were also highly expressed.

Oxytocin (*OXT*) was one of the most highly expressed genes in the CL. In cyclic ewes, *OXT* secreted from the CL and posterior pituitary is widely known to bind with oxytocin receptor (*OXTR*) from the endometrium to concomitantly release prostaglandin F_2α_ (*PGF*) pulses and induce luteolysis [15,23,50,51]. However, for noncyclic ewes, *OXT* plays an important role during peri-implantation and throughout pregnancy [52]. *OXT* signaling is known to be influenced by progesterone, but the mechanism underlying the regulation is not yet clear due to conflicting findings [53,54,55,56]. Meanwhile *OXTR* (TPM = 0.2) expression was almost negligible compared to OXT expression. *LOC114112617* marked as “uncharacterized ncRNA” is indeed a mitochondrial genome of sheep as blast search of its fasta sequence had 99.9% sequence similarity (query coverage of 100%) to mitochondrial genomes of several sheep breeds, including Finnsheep. Mitochondrial genes are typically abundantly expressed in high-energy demanding tissues and we observed similar expression in ovaries during follicular growth phase [13]. We identified several genes marked as “uncharacterized ncRNA” and those genes should instead be referred to as “uncharacterized contigs”. 

Two members of tissue-derived matrix metalloproteinase (*MMP*) inhibitors, also referred to as tissue inhibitors of metallopeptidases (TIMPs), were among the top 25 most abundant genes, of which *TIMP1* was ranked number 1 and *TIMP2*, number 24. In addition to *TIMP1* and *TIMP2*, at least 20 MMPs (including two isoforms of *MMP9* and *MMP24*) two TIMPs, *TIMP3* (TPM = 265) and *TIMP4* (TPM = 0.5) were expressed in our samples. The high level of TIMPs indicated successful pregnancies, as these inhibitors have low level of expression while their target MMPs are elevated during luteolysis [57,58,59]. TIMPs have an important role in the regulation of several processes relevant to uterine physiology including angiogenesis [60], cell differentiation [61], and embryo development [62]. Northern blot analyses in several tissues of sheep showed that both *TIMP1* and *TIMP2* had the greatest abundance in CL during early pregnancy [63]. Moreover, MMPs and TIMPs have important roles in implantation and that the high-level expression of *TIMP1* and *TIMP2* indicated active invasion of trophoblast cells during implantation [64,65]. Another study in bovine oviduct showed that *TIMP1* and *TIMP2* were highly expressed during the time of ovulation [66]. Hence, these two genes may play significant roles during the whole reproduction process—from ovulation to implantation. 

Translationally controlled tumor protein (*TCTP*) is a highly conserved, multifunctional protein that plays essential roles in development and other biological processes in different species [67,68,69,70,71]. With a maximum level of expression on Day 5 of pregnancy, this protein has been shown to play a significant role in embryo implantation in mice [68]. Consistent with these earlier studies, *TCTP* appeared to have the highest level of expression during the embryo implantation period. Matrix Gla protein (*MGP*) is a vitamin K-dependent extracellular matrix protein whose expression has been shown to be correlated with development and maturation processes [72,73] and receptor-mediated adhesion to the extracellular matrix [74]. Several studies have reported that *MGP* is highly expressed in the bovine endometrium [22,75,76] and we have shown that this is also the case with CL. The high level of expression of *MGP* in our study is consistent with the results of earlier studies in which this gene was found to be elevated during the preimplantation stage in sheep [25,76] and cattle [22]. Similarly, *MGP* was significantly upregulated in non-regressed compared to regressed bovine CLs [77]. Our data and supporting results from earlier studies on cattle show that *MGP* is highly expressed in CL during the preimplantation stage and plays important roles in superficial implantation and placentation in sheep. 

### 3.5. Breed Wise Gene Expression Differences in the CL

The highest number (n = 133) of differentially expressed genes were available for pairwise comparison involving the purebred Finnsheep and Texel (Figure 2, Appendix A). Similarly, 31 genes were differentially expressed between Finnsheep and F1 crossbred ewes, of which 19 genes were upregulated in Finnsheep (Appendix A). The lowest number (n = 27) of DEGs was observed between Texel and F1 crossbred ewes, with two-third genes being upregulated in F1 (Appendix A).

Out of the 133 significant DEGs in the CL of pure breeds (i.e., Finnsheep *vs.* Texel), 90 were upregulated in Finnsheep (Appendix A) and the rest were downregulated. About 10% of the DEGs were uncharacterized ncRNAs indicating the role of non-coding (preferably, long noncoding) RNAs in breed-wise difference during early pregnancy. However, blast search against the non-redundant (NR) database showed that these ncRNAs are possibly contigs spanning one or more genes. In the list of DEGs, we observed a few cases in which more than one gene from the same family was present. Four isoforms of multidrug resistance-associated protein 4-like (*MRP4*) were upregulated in Texel ewes. Earlier reports have suggested a role of *MRP4* in transporting prostaglandins in the endometrium [78], and *MRP4* has been found to be upregulated in the endometrium in infertile cows compared to fertile cows [27]. Although there are no reports regarding the existence and roles of *MRP4* in the CL, we speculate that the comparatively lower levels of these prostaglandin (PG) transporters in Finnsheep provide a luteoprotective effect. Six sialic acid-binding Ig-like lectins (Siglecs) including three isoforms of *SIGLEC-14* were upregulated in Finnsheep. Siglecs are transmembrane molecules expressed on immune cells and mediate inhibitory signaling [79]. So far, *SIGLEC-13* has been reported only in nonhuman primates; it was deleted during the course of human evolution [80]. The importance of Siglecs in immune system regulation has been reviewed elsewhere [81]. Siglecs constantly evolve through gene duplication events and may vary between species and even within a species [81,82,83]. Here, we have reported the expression of Siglecs in CL, which are known to play a role in the immune response during early pregnancy (preimplantation). 

Several cytokines including four chemokines (*CCL5*, *CCL24*, *CXCL9*, and *CXCL13*), two interleukin receptors (*IL1RN*, *IL12RB1*) and two interferons (*ISG20*, *GVINP1*) were upregulated in Finnsheep. Other genes with more than one member included major histocompatibility complexes (MHCs) (*HA1B*, *MICB*, *MICA*, *BOLA-DQB*0101*, etc.), tripartite motif-containing proteins (two isoforms of *TRIM5*, and *TRIM10*), cluster of differentiation factors (*CD4*, *CD69*), CD300 family molecules (*CD300C*, and *CD300H*). Several of the upregulated genes including cytokines and MHCs indicated an enhanced immune system during early pregnancy in Finnsheep compared to Texel. A novel protein coding gene (*LOC114116052*) was significantly upregulated in Texel. Nucleotide blast search of this novel gene’s CDS revealed its sequence similarity with *SENP6* in several mammalian species including transcript variants of sheep *SENP6* (*LOC101118793*, 95% query coverage and 98.7% sequence identity). Therefore, we concluded that *LOC114116052* is one of the isoforms of *SENP6*. All four ribosomal proteins (*MRPS33*, *RPS3A*, *RPL17*, and *RPL34*) differentially expressed between the pure breeds were upregulated in Finnsheep. Similarly, 16 of the differentially expressed genes were classified as uncharacterized ncRNAs, of which nine were upregulated in Finnsheep. It should be noted, however, that these uncharacterized ncRNAs might include coding genes and thus need further annotation curation. 

Out of 90 genes that were significantly upregulated in Finnsheep compared to Texel, 62 were recognized by ClueGO. However, 22 of the recognized genes lacked functional annotation. Thus, enrichment analysis for GO terms and KEGG pathways is based on 40 genes. In the end, only 16 genes passed the selection criteria (see Materials and Methods section) and were associated with 16 representative terms and pathways (Figure 3, Appendix A). It turned out that several of the terms and pathways comprised similar genes and grouping based on common genes revealed four different types of terms and/or pathways. The enriched terms and pathways belonged to four main categories based on the way genes were shared. The representative terms and/or pathways associated with upregulated genes were “negative regulation of viral life cycle”, “Th1 and Th2 cell differentiation”, “chemokine receptor binding” and, “dicarboxylic acid transport”. In summary, genes involved in the immune response were upregulated in Finnsheep CL during early pregnancy. Immune-related processes were mainly upregulated in the endometrium of prolific Meishan pig compared to Yorkshire [85]. We speculate that immunity plays bigger role in high prolific Finnsheep compared to Texel during preimplantation.

### 3.6. Uniquely Differentially Expressed Genes

We noticed that several genes (see Figure 4) were differentially expressed in more than one comparison, increasing our confidence in the identification of these DEGs. Few DEGs were exclusively up- or downregulated in one breed compared to the other two. *ANOS1*, *CLVS2*, *FOLR3*, *EEF1A1* and *LOC114115287* (uncharacterized ncRNA) were always upregulated in Finnsheep, whereas seven genes were exclusively downregulated. Although *LOC114115287* has been marked as an uncharacterized ncRNA, blast search revealed that more than 99% of its sequence is similar to the sheep mitochondrial genome. This is just another example showing that the gene level annotation of sheep reference genome still needs significant improvement. *MEPE* also was found to be downregulated in Finnsheep during ovulation [13]. *SENP6*, a protease involved in depolymerization of the small ubiquitin-like modifiers (SUMO-1/2) [86], was the most downregulated gene in Finnsheep compared to both Texel (log2FoldChange = −10.7, padj = 1.07 × 10^−7^, Appendix A) and F1(log2FoldChange = −10.2, padj = 1.07 × 10^−6^, Appendix A). A recent study reported that loss of *SENP6* in adult mice led to early embryonic death as well as postnatal premature ageing with skeletal abnormalities [87]. Moreover, knockdown of *SENP6* was also linked to upregulated expression of proinflammatory genes [88]. In agreement with our phenotype results (see Table 1) and earlier findings, we speculate that low *SENP6* gene expression in Finnsheep is linked to high embryo mortality, and vice versa in the case of Texel. 

Seven genes (*DNER*, *LOC106990106* (*EZR*), *LOC114109364* (*H3.3*), *LOC114115573* (*COX7C*), *LOC114115623* (*TOMM20*) and *LOC114115666* (*EEF2*)) were exclusively upregulated and 13 genes, including three ncRNAs, were exclusively downregulated in Texel (Figure 4). *TXNL1* was the most downregulated gene in Texel compared to both Finnsheep (log2FoldChange = −9.8, padj = 6.4 × 10^−21^, Appendix A) and F1 (log2FoldChange = −8.8, padj = 4.8 × 10^−16^, Appendix A). Interestingly, *TXNL1* is one of the candidate genes associated with embryonic diapause. Embryonic diapause is an intriguing condition where the embryo development is suspended because of unfavorable conditions for implantation [89]. The environmental conditions, such as food availability, daylight and temperature, and endogenous factors resulting from the ovary, pituitary gland and endometrium have been linked with embryonic diapause [90]. Finnsheep have up to four waves of follicular growth, leading to multiple ovulations [91], and have an extended breeding season [92]. The lambs that were occasionally born any time of the year were thought to be due to accidental mating [93,94]. Based on recent findings on the possibility for embryonic diapause to exist across mammals and high level of expression of *TXNL1* in our data, Finnsheep have potential for embryonic diapause. In particular, an extended breeding season and out of season breeding might be associated with embryonic diapause. Identification and characterization of other candidate genes in relevant tissues including endometrium in an experimental setting also involving the ewes pregnant for out-of-season breeding is essential for further validation. In addition, similar high-level expression of *TXNL1* in F1 cross indicated parent-of-origin effect (POE).

Similarly, *CA5A*, *FGF23* and *LOC114109527* (*NCLP*) were exclusively upregulated in F1 and *STX7* was downregulated. Among these, *CA5A* appeared to be upregulated in F1 crosses compared to both Finnsheep and Texel from both phases (i.e., in the CL in this study and in the ovary in our earlier study). *CA5A* is a member of the carbonic anhydrase family of zinc-containing metalloenzymes, whose primary function is to catalyze the reversible conversion of carbon dioxide to bicarbonate. The mitochondrial enzyme *CA5A* plays an important role in supplying bicarbonate (HCO_3-_) to numerous other mitochondrial enzymes. In a previous study, we observed downregulation of *CA5A* in the ovaries of Texel compared to F1 [13]. More recently, *CA5A* was also shown to be expressed in the ovaries of the Pelibuey breed of sheep; the gene was upregulated in a subset of ewes that gave birth to two lambs compared to uniparous animals [95]. However, there are no reports regarding the expression and function of *CA5A* in the CL. Based on the results from our current and earlier reports [13,95], *CA5A* appears to have an important function, at least until the preimplantation stage of reproduction. The level of expression in F1 crosses in the CL followed the same pattern as that in the ovary, which led us to conclude that *CA5A* is heritable and potentially an imprinted gene. Further experiments are needed to determine whether the gene is associated with high prolificacy.

### 3.7. Influence of Flushing in Gene Expression

Altogether, 19 DEGs including three uncharacterized non-coding RNAs were influenced by the flushing diet. Interestingly, the highest (n = 12) number of DEGs was present between the Finnsheep and F1 crosses and the lowest (n = 4) number was between the pure breeds (Table 4). Four genes were significantly differentially expressed between Finnsheep and Texel of which three were upregulated in Texel. The only gene upregulated in Finnsheep was cytochrome c oxidase subunit 7c, mitochondrial (*COX7C*). The three genes significantly upregulated in Texel were *PAPPA2*, *LOC114116073* (ncRNA, uncharacterized) and *LOC114118704* (Golgi apparatus membrane protein *TVP23* homolog B pseudogene). Out of 12 genes differentially expressed between Finnsheep and F1, five were upregulated in Finnsheep, while seven were upregulated in F1. *TVP23* pseudogene was always downregulated in Finnsheep as a result of the flushing diet. Similarly, *HOXD10*, *LOC105611004* (ncRNA, uncharacterized) and *LOC114116073* were upregulated in F1 compared to Finnsheep. 

An uncharacterized ncRNA (*LOC114116073)* was strongly downregulated in Finnsheep compared to both Texel (LFC = −4.6, padj = 1.26 × 10^−28^) and F1(LFC = −4.82, padj = 5.58 × 10^−31^). The 13,771 nucleotide-long fasta sequence of this uncharacterized ncRNA was queried to nucleotide blast search against NR database. Among the hits, we noticed at least two genes, *ULBP1* and *NKG2D*, among which *ULBP1* binds with *NKG2D* to trigger an immune response [96]. In conclusion, the flushing diet in Texel and F1 crosses led to improved immune surveillance mediated by *ULBP1* and *NKG2D*. Three subsets of homeobox (Hox) genes were upregulated in F1 as a result of flushing diet. *HOXD10* and *HOXA9* were upregulated against Finnsheep and Texel, respectively, and *HOXC11* was upregulated compared to both purebreds. Hox genes are transcriptional factors that precisely regulate embryo development as well as postnatal life. Hox genes are also considered as master regulators due to their role in directing the formation of many body parts. The developmental importance of Hox genes are reviewed elsewhere [97,98,99]. 

### 3.8. Limitations and Thoughts for Future Studies

We acknowledge certain limitations of this study. With sequencing costs becoming increasingly inexpensive, increasing the sample size of each breed group would certainly add statistical power. Given that time-series experiments are not feasible with the same animal, sampling could be performed with a larger group of animals at different stages of pregnancy to obtain an overview of gene expression changes. The newer Rambouillet reference genome worked better with our data (in terms of alignment and gene expression) compared to Oarv3.1 (data not shown). However, several contigs that are currently marked as “uncharacterized ncRNAs” need to be characterized before the reference genome can be fully utilized. One ovary from each ewe was removed earlier [13] and all the CLs for this study were collected from the remaining ovary. Therefore, there might be some impact due to possible negative feedback effects on overall gene expression. It should be noted that overall gene expression and, more specifically, differential expression between breeds is inherently a stochastic process; thus, there is always some level of bias caused by individual variation [100]. The results from breeding experiments show that productivity traits such as litter sizes may not carry on to F2 crosses (F1 × F1) and/or backcrosses. Therefore, future experiments that involve F2 crosses and backcrosses would provide more valuable findings related to prolificacy. Moreover, by doing a reciprocal cross experiment, we might be able to get insight into parent-of-origin effect (POE) and measure the potential contribution of the Texel and Finnsheep in each cross. In addition, replicating such experiments in different environments would be relevant for breeding strategies to mitigate the effects of climate change. To minimize or alleviate noise from tissue heterogeneity, single-cell experiments may prove beneficial in future studies. 

## 4. Conclusions

Our phenotypic records indicate lower embryo survival rate in Finnsheep compared to Texel. The relative scarcity of transcriptomic information about the CL means that its functional importance is underrated. We identified several key transcripts, including coding genes (producing mRNA) and noncoding genes, that are essential during early pregnancy. Functional analysis primarily based on literature searches and earlier studies revealed the significant roles of the most highly expressed genes in pregnancy recognition, implantation and placentation. F1 crosses were more closely related to Finnsheep than to Texel, as indicated by phenotypic and gene expression results that need to be validated with additional experiments (with F2 and backcrosses). Several genes with potential importance during early pregnancy (including *SIGLEC14*, *SIGLEC8*, *TXNL1*, *MRP4,* and *CA5A*) were reported in the CL for the first time in any species. The results from this study show the importance of the immune system during early pregnancy and may even have greater significance to high prolificacy as revealed by significant upregulation of immune-related genes in Finnsheep. Flushing appeared to enhance the immune system and may influence embryo survival. Our results indicate potential for embryonic diapause in sheep and might have an important connection with extended and/or out-of-season breeding in certain breeds of sheep including Finnsheep. We also highlight the need for improved annotation of the sheep genome and emphasize that our data will certainly contribute to such improvement. Taken together, our data provide new information to aid in understanding the complex reproductive events during the preimplantation period in sheep and may also have implications for other ruminants (such as goats and cattle) and mammals, including humans. 

## Figures and Tables

**Figure 1 genes-11-00415-f001:**
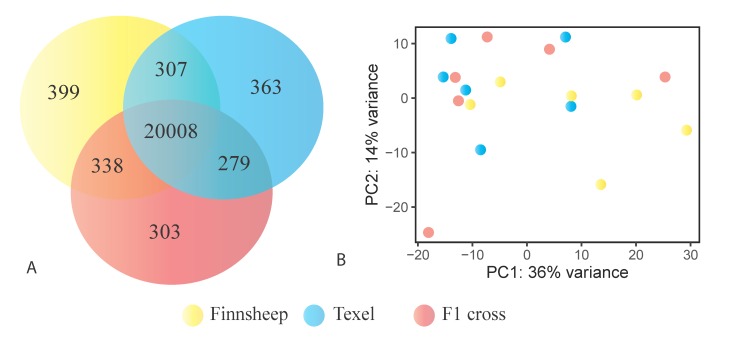
Graphical summary of gene expression in the *corpus luteum* (CL). (**A**) Shared and unique genes expressed in the CL of Finnsheep, Texel and their F1. (**B**) Principal component analysis (PCA) plot based on variance stabilized transformation (VST) of gene expression counts derived from DESeq2 in three breeds.

**Figure 2 genes-11-00415-f002:**
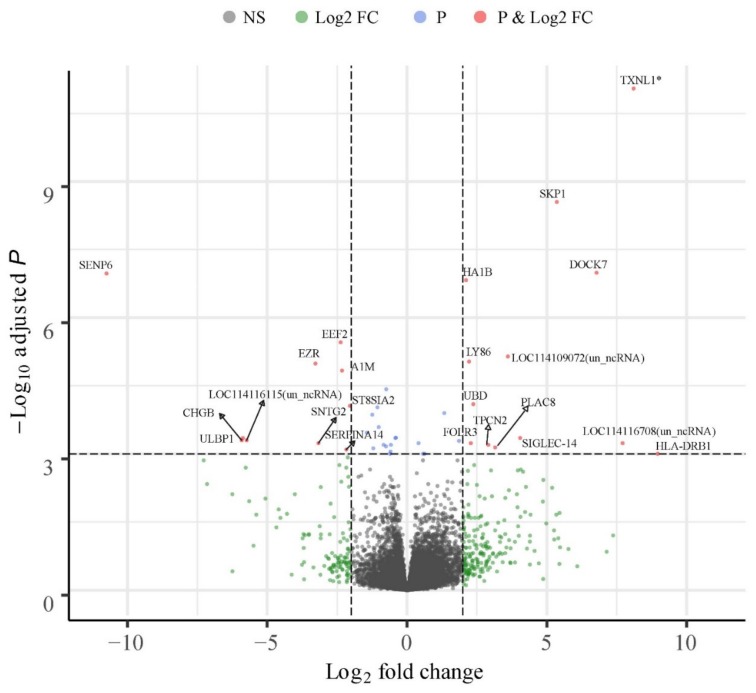
Volcano plot of 23,327 genes expressed in the pure breeds. Top 23 of 133 genes significantly differentially expressed between Finnsheep and Texel are denoted by gene names except three unknown ncRNAs. All DE genes with absolute log2 fold change greater than 2 and adjusted P value lower than 0.001 have been highlighted here. For additional details and complete list of 133 DE genes, please refer Appendix A. Note that *TXNL1* gene (marked by an asterisk) has an adjusted P-value of 4.84E-16. Volcano plot created using Bioconductor package EnhancedVolcano [84] and some adjustments in the figure were made in Adobe Illustrator (Adobe Inc.). The x-axis represents log2 fold change and the y-axis represents adjusted *p*-values (−log10). Legends in top: NS = non-significant; Log2 FC = genes having absolute log2 fold change greater than 2; P = genes with adjusted *P* value less than 0.001; and P & Log2 FC: genes passing the criteria for log2 fold change and adjusted *P* value.

**Figure 3 genes-11-00415-f003:**
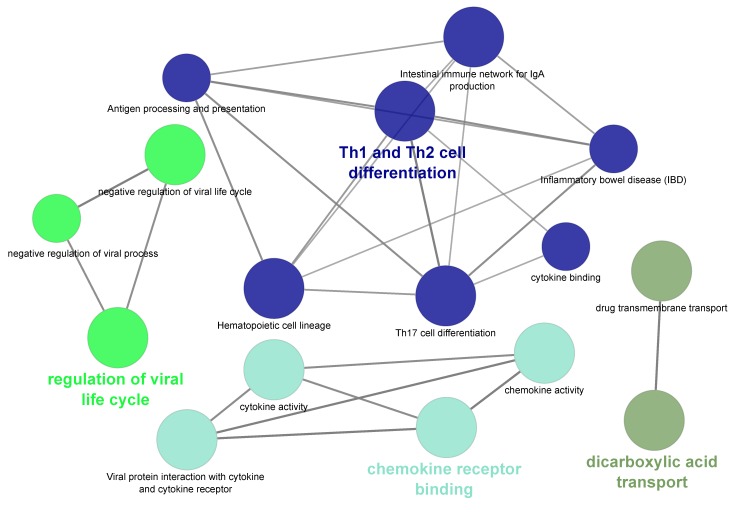
Gene Ontology (GO) terms and Kyoto Encyclopedia of Genes and Genomes (KEGG) pathways associated with significantly differentially expressed genes between Finnsheep and Texel. The functional grouping option in ClueGO categorized 16 GO terms and pathways into four groups using kappa score where the genes shared between the terms are iteratively compared to form functional groups.

**Figure 4 genes-11-00415-f004:**
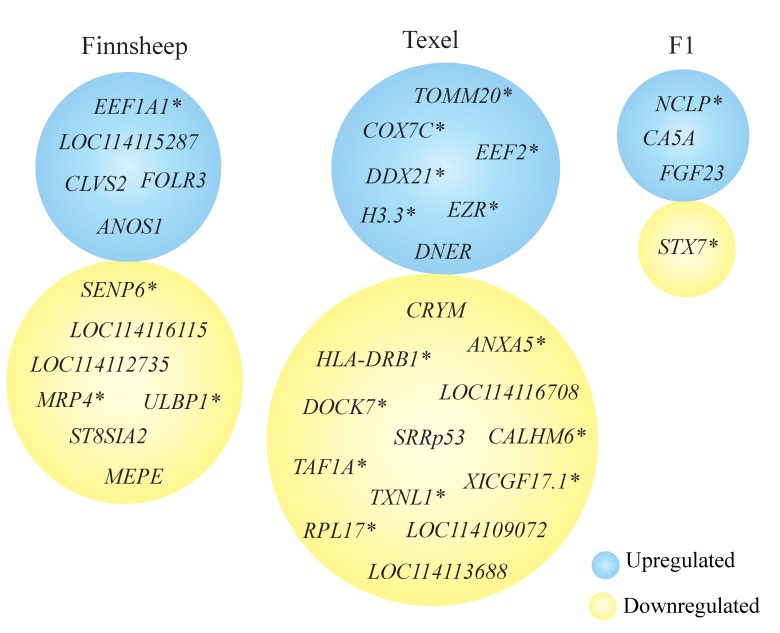
List of uniquely differentially expressed genes in Finnsheep, Texel and F1. Gene names that were not part of Rambouillet reference transcriptome are marked with an asterisk (*) while gene IDs starting with LOC are “uncharacterized ncRNA”.

**Table 1 genes-11-00415-t001:** Average number of *corpus luteum* (CL) and embryos in Finnsheep, Texel and their F1 crosses. The full details of the phenotypes are available in Appendix A.

Breed	# CL	# Embryos
Finnsheep	4.1	2.6
Texel	1.7	1.5
F1	3.75	1.8

**Table 2 genes-11-00415-t002:** Sample summary. A total of 21 samples with 7 each for Finnsheep (FS), Texel (TX) and F1 crosses of Finnsheep and Texel (F1) were included in this study. Diet is either normal diet (C) or flushing diet (F). Alignment statistics from STAR include only the uniquely aligned reads whereas those from Salmon include all reads that mapped to reference transcriptome. Numbers for sequence reads are represented as millions (M).

Sample	Breed	Diet	Raw Reads (M)	Clean Reads (M)	Uniquely Aligned Reads (STAR, M)	Aligned Reads (Salmon, M)	ENA Accession Code
1033	TX	C	56.4	55.9	48.0	50.7	ERR3349023
107A	TX	F	55.0	54.5	46.5	49.3	ERR3349025
107B	TX	F	55.7	55.2	48.3	49.8	ERR3349027
251	TX	F	41.8	41.2	35.4	36.9	ERR3349029
302	TX	C	54.5	54.1	47.1	48.5	ERR3349031
312A	FS	F	54.9	54.4	47.0	49.3	ERR3349033
312B	FS	F	79.2	78.4	69.2	71.1	ERR3349035
3609	FS	C	54.5	54.1	47.0	49.0	ERR3349037
379	TX	F	43.1	42.7	36.0	38.3	ERR3349039
4208	F1	C	54.3	53.8	44.8	49.0	ERR3349041
4271A	F1	F	47.2	46.8	40.1	42.4	ERR3349043
4271B	F1	F	46.9	46.4	40.0	42.1	ERR3349045
4519	F1	F	51.6	51.1	44.5	46.3	ERR3349047
4563	F1	F	54.1	53.7	45.7	48.6	ERR3349049
4590	F1	F	48.5	48.1	40.0	43.5	ERR3349051
4823	F1	C	52.6	52.1	42.6	47.2	ERR3349053
48	FS	C	53.4	53.0	44.7	47.9	ERR3349055
554	FS	F	56.0	55.5	48.1	49.7	ERR3349057
73	TX	C	46.1	45.7	39.1	41.3	ERR3349059
897	FS	F	53.3	52.9	46.1	47.9	ERR3349061
974	FS	C	45.1	44.7	38.6	40.4	ERR3349063

**Table 3 genes-11-00415-t003:** List of top 25 most abundant genes expressed in the CL. Gene names and gene descriptions for gene IDs starting with “LOC” were retrieved from NCBI Genbank (https://www.ncbi.nlm.nih.gov/genbank/) and, GeneCards (https://www.genecards.org/), respectively. *LOC114112617* marked originally as “uncharacterized ncRNA” in NCBI genbank, is indeed the mitochondrial genome of sheep.

Gene ID	Gene Description	Chr	Mean TPM
*TIMP1*	Tissue inhibitor of metallopeptidase 1	X	16705.6
*STAR*	Steroidogenic acute regulatory protein	26	14226.6
*OXT*	oxytocin/neurophysin I prepropeptide	13	13113.2
*MGP*	matrix Gla protein	3	12294.0
*LOC114112617*	Sheep mitochondrial genome	MT	12155.9
*LOC114113966*	translationally controlled tumour protein (*TCTP*)	3	8341.7
*HSD3B1*	hydroxy-delta-5-steroid dehydrogenase, 3 beta- and steroid delta-isomerase 1	1	8249.3
*LOC101117785*	Apolipoprotein A1 (*APOA1*)	15	8213.2
*LOC101110773*	elongation factor 1-alpha 1 (*EEF1A1*)	10	7564.5
*LOC101103639*	uncharacterized (protein coding)	3	7272.8
*LOC114116158*	thymosin beta-4 pseudogene	1	6689.3
*LOC114118103*	40S ribosomal protein S29	14	5599.0
*CYP11A1*	Cytochrome P450 Family 11 Subfamily A Member 1	18	5096.8
*RPLP1*	ribosomal protein lateral stalk subunit P1	7	5032.2
*LOC114116632*	60S ribosomal protein L23a (*RPL23a*)	10	4998.9
*FTH1*	Ferritin heavy chain 1	21	4498.0
*RPS8*	ribosomal protein S8	1	4402.1
*LOC105606567*	60S ribosomal protein L39	4	4215.9
*MSMB*	microseminoprotein beta	25	4093.6
*RPS24*	ribosomal protein S24	13	4029.1
*RPS17*	ribosomal protein S17	18	3950.0
*LOC114108766*	60S ribosomal protein L17 (*RPL17*)	4	3832.5
*RPS18*	ribosomal protein S18	20	3657.2
*TIMP2*	Tissue inhibitor of metallopeptidase 2	11	3587.4
*LOC114114908*	40S ribosomal protein S27 (*RPS27*)	5	3570.4

TPM: Transcripts Per Million, X: chromosome X, MT: Mitochondrial genome.

**Table 4 genes-11-00415-t004:** List of genes differentially expressed because of the flushing diet in Finnsheep (FS), Texel (TX) and F1 crosses (F1). Gene IDs lacking gene name and/or descriptions (marked with an *) were checked again in the NCBI to retrieve additional information.

Gene	Gene *	Base Mean	LFC	Padj	Condition
*COX7C*		101.753584	1.93	0.03584019	FS vs. TX
*LOC114116073*	un_ncRNA	832.4	−4.60	1.26 × 10^−28^	FS vs. TX
*PAPPA2*		24.4	−2.27	0.00091	FS vs. TX
*LOC114118704*	TVP23(p)	38.8	−2.51	1.70 × 10^−6^	FS vs. TX
*ADAMTS4*		892.8	−1.80	0.02409534	FS vs. F1
*SHOX2*		74.0	1.95	0.0200087	FS vs. F1
*HOXD10*		99.3	−2.17	0.00279421	FS vs. F1
*SCTR*		87.9	1.70	0.0146455	FS vs. F1
*HOXC11*		24.1	−1.81	0.01918401	FS vs. F1
*LOC105611004*	un_ncRNA	40.9	−2.06	0.00279421	FS vs. F1
*LOC114116073*	un_ncRNA	832.4	−4.82	5.58 × 10^−31^	FS vs. F1
*LRP11*		125.0	−1.84	0.02732988	FS vs. F1
*LOC114118704*	TVP23(p)	38.8	−1.92	0.00279421	FS vs. F1
*BTN1A1*		22.4	1.86	0.02926403	FS vs. F1
*KCNQ1*		857.0	1.77	0.04977007	FS vs. F1
*HNR1PA1*		81.1	1.93	0.0200087	FS vs. F1
*COX7C*		101.7	−1.88	0.03094773	TX vs. F1
*PCSK5*		469.2	1.69	0.01712337	TX vs. F1
*SCTR*		87.9	1.79	0.00816462	TX vs. F1
*HOXC11*		24.1	−1.92	0.00816462	TX vs. F1
*PPM1H*		575.3	1.67	0.00684851	TX vs. F1
*HOXA9*		74.2	−1.96	0.00816462	TX vs. F1
*RDH10*		2603.9	1.56	0.00816462	TX vs. F1
*CA5A*		983.4	−1.50	0.01137482	TX vs. F1

LFC: Log2(Fold Change), Padj: Adjusted p-value.

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
