# Peer review of "Gene Expression Profiling of Corpus luteum Reveals Important Insights about Early Pregnancy in Domestic Sheep"

_genes, 2020, doi:10.3390/genes11040415_

Round 1

Reviewer 1 Report

Very well-prepared manuscript and it appears to be the second manuscript in a series. The first manuscript was published in 2018 in BMC Genomics. It appears these authors were very skilled at the processes of analyzing and presenting RNA-Seq/gene expression data; however,  there were three primary issues that need clarifications: 

  1. The overall experimental design appears to have main effects of breed, diet, and the interaction of breed x diet. This is a lengthy information-dense manuscript. Perhaps this information was stated, but it was unclear to this reviewer how these effects were accounted in the statistical analyses. 
  2. The phrase, "early pregnancy", was used frequently. What days are included in this phrase (12 to 18)? Furthermore, does the day of conception influence these results?
  3. Embryo mortality, diapause, vs health embryo are concerns about how the the animals were grouped in this study and the data presented. How do the investigators know that the animals sampled were in any of these group?

Other specific statement used through out this manuscript that can use improvement include:

  1. Sheep and the word "litter(s)" is a bit awkward, even when describing Finn Sheep. Perhaps other phrase-ology about lamb crop size is more appropriate. 
  2. Cliches' appear throughout the manuscript, such as: dataset, In line, etc. Concise writing can improve the manuscript. 
  3. Lines 67-68, "keeping half the of the ewes on flushing" is an awkward phrase. What does this mean? Only the tissues that were harvested from the ewes consuming the flush (i.e., high energy) diet were used in this study?

Reviewer 2 Report

The aim of this study was to identify the gene expression of corpus luteum and to compare its profile with the important events involved in embryo survival and pregnant rate on sheep. Using high prolific and low prolific breeds, the authors identity several key transcripts, including coding genes and noncoding genes, that are essential during early pregnancy because their implication in pregnancy recognition, implantation and placentation. Some of them were reported in the corpus luteum for the very first time in any species. Transcriptomics is a powerful tool for understanding the intimate mechanisms that take place in  the key tissues responsible for fertility. The experiments are well designed and executed, with the results clearly presented and concise and the discussions reveal the originality of paper results.

Author Response

Thank you very much for carefully evaluating our manuscript.